# Transplacental transmission of SARS-CoV-2 infection

Alexandre J. Vivanti 1,8, Christelle Vauloup-Fellous2,8, Sophie Prevot3, Veronique Zupan4, Cecile Suffee5, Jeremy Do Cao 6, Alexandra Benachi 1 & Daniele De Luca 4,7✉

SARS-CoV-2 outbreak is the first pandemic of the century. SARS-CoV-2 infection is transmitted through droplets; other transmission routes are hypothesized but not confirmed. So far, it is unclear whether and how SARS-CoV-2 can be transmitted from the mother to the fetus. We demonstrate the transplacental transmission of SARS-CoV-2 in a neonate born to a mother infected in the last trimester and presenting with neurological compromise. The transmission is confirmed by comprehensive virological and pathological investigations. In detail, SARS-CoV-2 causes: (1) maternal viremia, (2) placental infection demonstrated by immunohistochemistry and very high viral load; placental inflammation, as shown by histological examination and immunohistochemistry, and (3) neonatal viremia following placental infection. The neonate is studied clinically, through imaging, and followed up. The neonate presented with neurological manifestations, similar to those described in adult patients.

[1] Division of Obstetrics and Gynecology, Antoine Béclère Hospital, Paris Saclay University Hospitals, APHP 157 rue de la Porte de Trivaux, 92140 Clamart, France. [2] Division of Virology, Paul Brousse Hospital, Paris Saclay University Hospitals, APHP 12 Avenue Paul Vaillant Couturier, 94800 Villejuif, France. [3] Division of Pathology, Bicetre Hospital, Paris Saclay University Hospitals, APHP, Le Kremlin-Bicêtre, France. [4] Division of Pediatrics and Neonatal Critical Care, Antoine Béclère Hospital, Paris Saclay University Hospitals, APHP 157 rue de la Porte de Trivaux, 92140 Clamart, France. [5] Division of Radiology, Antoine Béclère Hospital, Paris Saclay University Hospitals, APHP 157 rue de la Porte de Trivaux, 92140 Clamart, France. [6] Division of General Pediatrics, Antoine Béclère Hospital, Paris Saclay University Hospitals, APHP 157 rue de la Porte de Trivaux, 92140 Clamart, France. [7] Physiopathology and Therapeutic Innovation Unit-INSERM U999, Paris Saclay University, 63 Rue Gabriel Péri, 94270 Le Kremlin-Bicêtre, France. [8]These authors contributed equally: Alexandre J. Vivanti, Christelle Vauloup-Fellous. ✉email: dm.deluca@icloud.com

SARS-CoV-2 infection causes the new coronavirus disease (COVID-19) and is mainly transmitted through droplets, but other transmission routes have been hypothesized. Some cases of perinatal transmission have been described[1–6], but it is unclear if these occurred via the transplacental or the transcervical route or through environmental exposure. It is important to clarify whether and how SARS-CoV-2 reaches the fetus, so as to prevent neonatal infection, optimize pregnancy management and eventually better understand SARS-CoV-2 biology. Here we present a comprehensive case study demonstrating the transplacental transmission of SARS-CoV-2 with clinical manifestation in the neonate, consistent with neurological signs and symptoms of COVID-19.

## Results

**Case study.** A 23-year-old, gravida 1, para 0 was admitted to our university hospital in March 2020 at $35^{+2}$ weeks of gestation with fever (38.6 °C) and severe cough and abundant expectoration since 2 days before hospitalisation. Real-time polymerase chain reaction (RT-PCR) was performed as described in the "Methods" below: both the E and S genes of SARS-CoV-2 were detected in blood, and in nasopharyngeal and vaginal swabs. Pregnancy was uneventful and all the ultrasound examinations and routine tests were normal until the diagnosis of COVID-19. Thrombocytopenia ($54 \times 10^9$/L), lymphopenia ($0.54 \times 10^9$/L), prolonged APTT (60 s), transaminitis

(AST 81 IU/L; ALT 41 IU/L), elevated C-reactive protein (37 mg/L) and ferritin (431 µg/L) were observed upon hospital admission. Three days after admission a category III-fetal heart rate tracing[7] (Fig. 1) was observed and therefore category II-cesarean section (i.e., fetal compromise; not immediately life-threatening, https://www.rcog.org.uk/globalassets/documents/guidelines/goodpractice11classificationofurgency.pdf) was performed, with intact amniotic membranes, in full isolation and under general anesthesia due to maternal respiratory symptoms. Clear amniotic fluid was collected prior to rupture of membranes, during cesarean section and tested positive for both the E and S genes of SARS-CoV-2. Delayed cord clamping was not performed as its effect on SARS-CoV-2 transmission is unknown. The woman remained hospitalized for surveillance of her clinical conditions and finally she was discharged in good conditions, 6 days after delivery.

A male neonate was delivered (gestational age $35^{+5}$ weeks; birth weight 2540 g). Apgar scores were 4 (in detail: heart rate = 1, respiratory activity = 1, skin color = 1, muscular tonus = 1, remaining items were coded zero), 2 (in detail: skin color = 1, muscular tonus = 1, remaining items were coded zero) and 7 (in detail: heart rate = 2, respiratory activity = 2, skin color = 2, muscular tonus = 1) at 1, 5 and 10 min, respectively. Neonatal resuscitation was provided according to current international guidelines[8] (face mask-delivered non-invasive ventilation from birth until 5 min of life and then intubation and invasive

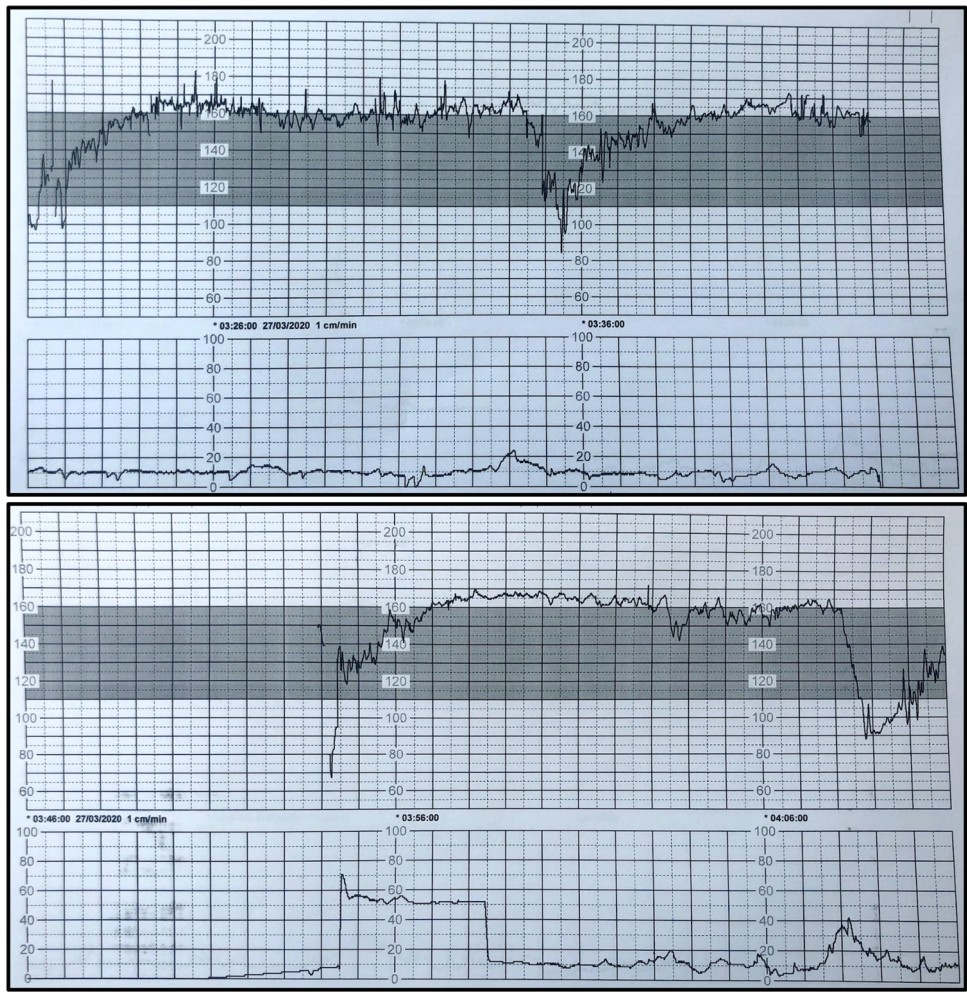

**Fig. 1 Illustrative snapshot of fetal heart rate tracing.** Tachycardia, absent baseline variability, absence of accelerations with recurrent prolonged and late decelerations. These findings are highly suggestive of a pathological category III fetal heart rate tracing[7], which is strongly associated with adverse neonatal outcome. This cardiotogram was recorded 26 min before the cesarean section.

ventilation with inspired oxygen fraction titrated up to 0.30; monitoring included ECG, end-tidal side-stream $CO_2$ measurements, peripheral oxygen saturation and perfusion index). The neonate was eventually transferred in full isolation to the neonatal intensive care unit (NICU) in a negative pressure room. Cord blood gas analysis showed normal pH and lactate. The neonate did not receive any sedative or analgesic drug and was monitored according to our routine NICU protocols for post-resuscitation care: Sarnat score, point-of-care echocardiography and lung ultrasound[9] were normal upon NICU admission. Vital parameters were always normal and the baby was extubated after ~6 h. Before the extubation, blood was drawn for capillary blood gas analysis (at 1.5 h of life) and routine blood tests, which yielded normal values. Moreover, before the extubation, blood and non-bronchoscopic bronchoalveolar lavage fluid were collected for RT-PCR and both were positive for the E and S genes of SARS-CoV-2. Lavage was performed using a standardized procedure[10] as detailed below. Blood culture was negative for bacteria or fungi. Nasopharyngeal and rectal swabs were first collected after having cleaned the baby at 1 h of life, and then repeated at 3 and 18 days of postnatal age: they were tested with RT-PCR and were all positive for the two SARS-CoV-2 genes. Routine blood tests (including troponin, liver and kidney function) were repeated on the second day of life and resulted normal. Feeding was provided exclusively using formula milk.

On the third day of life, the neonate suddenly presented with irritability, poor feeding, axial hypertonia and opisthotonos: cerebrospinal fluid (CSF) was negative for SARS-CoV-2, bacteria, fungi, enteroviruses, herpes simplex virus 1 and 2, showed normal glycorrhachia albeit with 300 leukocytes/mm$^3$ and slightly raised proteins (1.49 g/L). Blood was taken at the same time and the culture was sterile. Cerebral ultrasound and EEG were also normal. There were no signs suspected for metabolic diseases. Symptoms improved slowly over 3 days and a second CSF sample was normal on the fifth day of life, but mild hypotonia and feeding difficulty persisted. Main laboratory findings are resumed in Table 1. Magnetic resonance imaging at 11 days of life showed bilateral gliosis of the deep white periventricular and subcortical matter, with slightly left predominance (Fig. 2). The neonate did not receive antivirals or any other specific treatment, gradually recovered and was finally discharged from hospital after 18 days. Follow-up at almost 2 months of life showed a further improved neurological examination (improved hypertonia, normal motricity) and magnetic resonance imaging (reduced white matter injury); growth and rest of clinical exam were normal.

**Virology and pathology**. RT-PCR on the placenta was positive for both SARS-CoV-2 genes. Figure 3 shows all RT-PCR results obtained in different maternal and neonatal specimens: viral load was much higher in placental tissue, than in amniotic fluid and maternal or neonatal blood.

Placental histological examination was performed as described in "Methods" below and revealed diffuse peri-villous fibrin deposition with infarction and acute and chronic intervillositis. An intense cytoplasmic positivity of peri-villous trophoblastic cells was diffusely observed performing immunostaining with antibody against SARS-CoV-2 N-protein. No other pathogen agent was detected on special stains and immunohistochemistry. Figures 4 and 5 depict the results of the placental gross and microscopic examination, as well as immunohistochemistry.

## Discussion

We report a proven case of transplacental transmission of SARS-CoV-2 from a pregnant woman affected by COVID-19 during

**Table 1 Main laboratory findings in the neonate.**

|  | DOL1 | DOL2 | DOL3 | DOL5 |
|---|---|---|---|---|
| **Blood cell counts** |  |  |  |  |
| WBC/L | $10.32 \times 10^9$ | $6.97 \times 10^9$ |  |  |
| RBC/L | $4.54 \times 10^{12}$ | $4.84 \times 10^{12}$ |  |  |
| Hb (g/dL) | 13.9 | 14,7 |  |  |
| Hematocrit (%) | 41.6 | 41.4 |  |  |
| Platelets/L | $339 \times 10^9$ | $319 \times 10^9$ |  |  |
| Lymphocytes/L | $4.39 \times 10^9$ | $3.05 \times 10^9$ |  |  |
| Neutrophils/L | $3.97 \times 10^9$ | $2.78 \times 10^9$ |  |  |
| Reticulocytes (%) | 3.04 | 3.39 |  |  |
| **Blood gas analyses** |  |  |  |  |
| pH | 7.27 | 7.38 | 7.34 |  |
| pCO$_2$ (mmHg) | 41 | 41 | 47 |  |
| pO$_2$ (mmHg) | 41 | 40 | 30 |  |
| BE (mmol/L) | −7.8 | −1.4 | −1 |  |
| Lactate (mmol/L) | 7 | 1.3 | 1.5 |  |
| Na (mmol/L) | 135 | 141 | 141 |  |
| K (mmol/L) | 6.3 | 5.2 | 4.3 |  |
| Cl (mmol/L) | 109 | 110 | 110 |  |
| Ca$^{++}$ (mmol/L) | 1.08 | 1.33 | 1.38 |  |
| **Blood biochemistry** |  |  |  |  |
| CRP (mg/L) | <5 | <5 |  |  |
| PCT (µg/L) | 0.95 | 0.61 |  |  |
| Cord PCT (µg/L) | 0.19 |  |  |  |
| Total serum bilirubin (µmol/L) |  | 106 |  |  |
| Conjugated bilirubin (µmol/L) |  | 0 |  |  |
| Alkaline phosphatase (IU/L) |  | 133 |  |  |
| AST (IU/L) |  | 38 |  |  |
| ALT (IU/L) |  | 9 |  |  |
| γ-GT (IU/L) |  | 290 |  |  |
| Troponine I (ng/L) |  | 43 |  |  |
| **CSF analyses** |  |  |  |  |
| CSF proteins (g/L) |  |  | 1.49 | 1.24 |
| CSF leukocytes/mm$^3$ |  |  | 300 | 11 |
| CSF glucose (mmol/L) |  |  | 2.9 | 2.4 |

All samples have been obtained by venous puncture, except blood gas analyses that have been performed on arterialized capillary blood samples obtained by warmed heel prick. All measurements have been performed with certified analytical micro-methods dedicated to the NICU and subjected to periodic quality controls. All results are normal for the neonatal reference ranges, expect for CSF proteins and leukocytes on DOL3; glycorrhachia was always similar to blood glucose measured at the same time.
γ-GT, gamma-glutamyl transferase; ALT, alanine aminotransferase; AST, aspartate transaminase; BE, base excess; CRP, C-reactive protein; CSF, cerebro-spinal fluid; DOL, day of life; PCT, procalcitonin; RBC, red blood cells; WBC, white blood cells.

late pregnancy to her offspring. Other cases of potential perinatal transmission have recently been described, but presented several unaddressed issues. For instance, some failed to detect SARS-CoV-2 in neonates or only reported the presence of specific antibodies[1,2,4]; others found the virus in the newborn samples but the transmission route was not clear as placenta, amniotic fluid and maternal or newborn blood were not systematically tested in every mother-infant pair[3,5,6,11,12].

A classification for the case definition of SARS-CoV-2 infection in pregnant women, fetuses and neonates has recently been released and we suggest to follow it to characterize cases of potential perinatal SARS-CoV-2 transmission. According to this classification system, a neonatal congenital infection is considered proven if the virus is detected in the amniotic fluid collected prior to the rupture of membranes or in blood drawn early in life, so our case fully qualifies as congenitally transmitted SARS-CoV-2 infection, while the aforementioned cases would be classified as only possible or even unlikely[13]. Another recent report describes a case with similar placental findings, but it has been classified only as probable case of congenital SARS-CoV-2 infection, because cord and newborn blood could have not been tested[14].

Both "E" and "S" gene of SARS-CoV-2 were found in each and every specimen, thus they were considered all positive, according to the European Centre for Disease Control recommendations (https://www.ecdc.europa.eu/en/all-topics-z/coronavirus/threats-and-outbreaks/covid-19/laboratory-support/questions). Of note, the viral load is much higher in the placental tissue than in amniotic fluid or maternal blood: this suggests the presence of the virus in placental cells, which is consistent with findings of

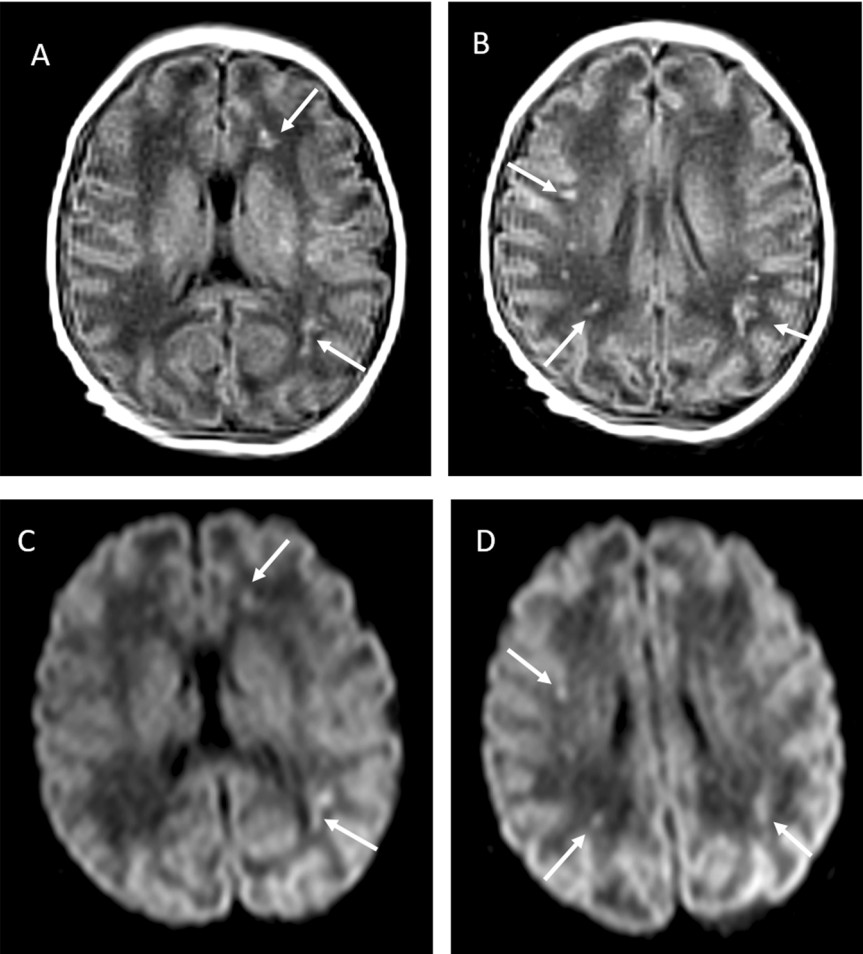

**Fig. 2 Cerebral MRI performed at 11 days of life. a**, **b** and **c**, **d** T1 and diffusion-weighted sequences, respectively. Images are taken at two different levels and show hyperintensities of the periventricular and subcortical frontal or parietal white matter (arrows).

inflammation seen at the histological examination. Finally, the RT-PCR curves of neonatal nasopharyngeal swabs at 3 and 18 day of life are higher than that at the first day (while the baby was in full isolation in a negative pressure room): this is also another confirmation that we observed an actual neonatal infection, rather than a contamination. Thus, these findings suggest that: (1) maternal viremia occurred and the virus reached the placenta as demonstrated by immunohistochemistry; (2) the virus is causing a significant inflammatory reaction as demonstrated by the very high viral load, the histological examination and the immuno-histochemistry; (3) neonatal viremia occurred following placental infection. Our findings are also consistent with a case study describing the presence of virions in placental tissue, although this did not report neither placental inflammation, nor fetal/neonatal infection[15].

The placenta showed signs of acute and chronic intervillous inflammation consistent with the severe systemic maternal inflammatory status triggered by SARS-CoV-2 infection. As RT-PCR on the placental tissue was positive for SARS-CoV-2, and both maternal and neonatal blood samples were also positive, the transmission clearly occurred through the placenta. Interestingly, placentas from women affected by SARS-CoV-1 presented similar pathological findings of intervillositis, with intervillous fibrin deposition[16]. Angiotensin-converting enzyme 2 (ACE2) is known to be the receptor of SARS-CoV-2 and is highly expressed in placental tissues[17]. Animal data show that ACE2 expression

changes in fetal/neonatal tissues over time and reaches a peak between the end of gestation and the first days of postnatal life[17]. The combination of these data and our findings confirms that transplacental transmission is indeed possible in the last weeks of pregnancy, although we cannot exclude a possible transmission and fetal consequences earlier during the pregnancy, as there are no definite literature data available yet.

Interestingly, we described a case of congenital infection associated with neurological manifestations following neonatal viremia. Suspected neonatal SARS-CoV-2 infections presented with non-specific symptoms[4] or pneumonia[3], while neurological symptoms are commonly observed in adult patients, especially due to the inflammatory response[18,19]. Early neurological mani-festations were also observed in another neonate born to SARS-CoV-2 positive mother, although vertical transmission was not fully investigated[12]. Conversely, after the viremia, our case clearly presented neurological symptoms and inflammatory findings in CSF. There was no other viral or bacterial infection and all other neonatal disorders potentially causing these clinical manifesta-tions were excluded. Neuroimaging consistently indicated white matter injury, which can be caused by the vascular inflammation induced by SARS-CoV-2 infection, as similar images have been anecdotally found in adult patients[20,21].

In conclusion, we have demonstrated that the transplacental transmission of SARS-CoV-2 infection is possible during the last weeks of pregnancy. Transplacental transmission may cause

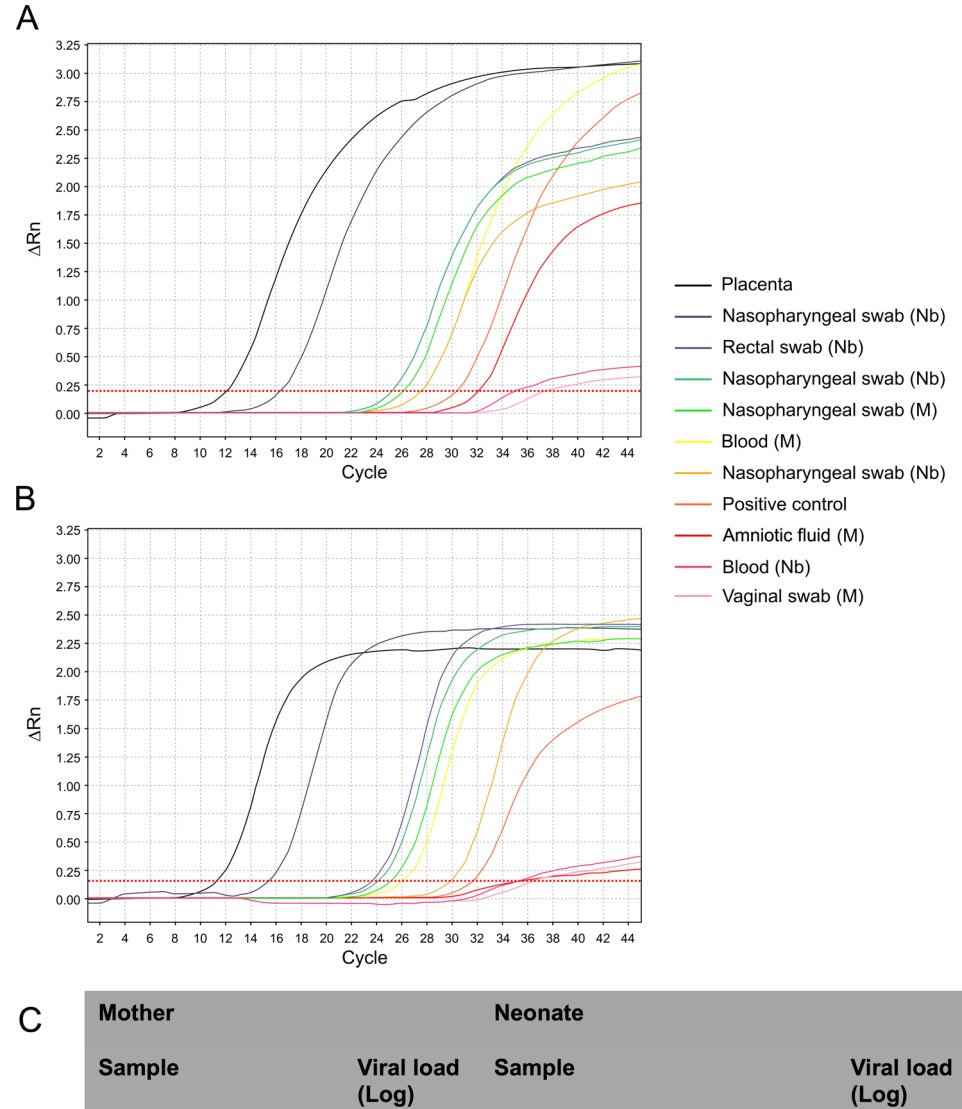

**Fig. 3 Real-time polymerase chain reaction results. a**, **b** The E and S genes of SARS-CoV-2, respectively, for maternal and neonatal samples (*X* and *Y* axes represent the amount of amplified RNA and the number of cycles, respectively; the earlier the signal is detected, the lowest is the number of cycles and the higher the viral load is). **c** The viral load for each sample (expressed as Log copies/million of cells for the placenta and as Log copies/mL for all other specimens). All maternal samples were obtained right before the delivery or during C-section; newborn samples are listed chronologically and were obtained from the first to the third day of life, except for the last nasopharyngeal swab (obtained at 18 days of postnatal age). Colored lines represent the results of RT-PCR assay for each sample. The deep orange line represents the positive control, which is a SARS-CoV-2 culture supernatant (more details in "Methods"). Nasopharyngeal swabs at 1, 3 and 18 day of life are represented by the light orange, gray and green curves, respectively. Viral load in BAL fluidis not shown. DOL days of life, M maternal samples, Nb newborn samples.

placental inflammation and neonatal viremia. Neurological symptoms due to cerebral vasculitis may also be associated.

## Methods

**Patient sampling**. Biological samples to be tested by RT-PCR were obtained and prepared as follows. Nasopharyngeal and vaginal swabs were obtained following US Center for Disease Control and Prevention guidelines (https://www.cdc.gov/coronavirus/2019-ncov/hcp/inpatient-obstetric-healthcare-guidance.html; https://www.cdc.gov/groupbstrep/downloads/gbs_swab_sheet21.pdf). A sample of placental tissues was taken from the chorionic side and crushed in 400 mL of RNAase-DNAase-free water; 1 mL of blood and swabs were placed in Virocult® viral transport media (Sigma, St. Louis, MI, USA). Non-bronchoscopic bronchoalveolar lavage (BAL) was performed following a well-known standardized technique[10]: in

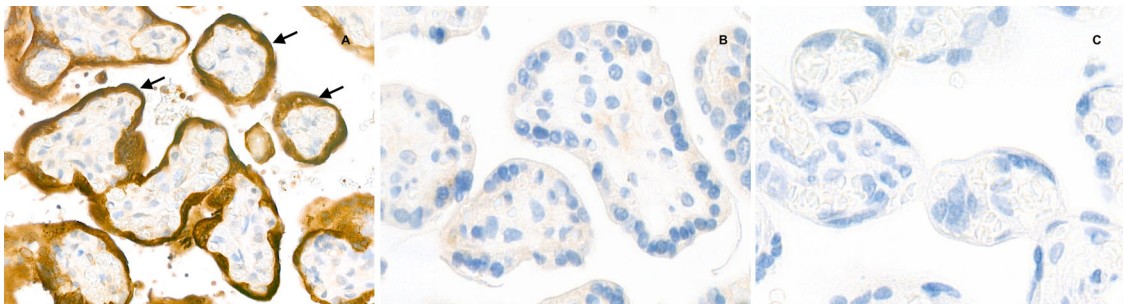

**Fig. 4 Gross and microscopic examination of the placenta. a** The macroscopic lesions of perivillous fibrin deposition with infarction, as irregular strands of pale yellow-white induration (arrow). **b** Microscopic lesions of intervillositis characterized by an infiltrate of the intervillous spaces made of neutrophils and histiocytes (arrow) (HES stain, original magnification ×400). **c** The intervillositis with several CD68-positive histiocytes (arrow); neutrophils are negative with this anti-macrophage antibody (anti-CD68 immunohistochemistry, original magnification ×400).

**Fig. 5 Placental immunostaining for SARS-CoV-2 N-protein (anti-N immunohistochemistry, original magnification ×800). a** The intense brown cytoplasmic positivity of peri-villous trophoblastic cells in the placenta of our case (arrows). **b, c** Two negative controls (primary antibody, two SARS-CoV-2 negative placentas).

detail, the neonate was placed supine with the head turned to the right so that the left lung would be predominantly sampled. Normal saline (1 mL/kg, 37 °C) was instilled into the endotracheal tube through a Y-piece. After three ventilator cycles, the suction catheter was gently inserted 0.5 cm beyond the tube tip, and the airway fluid was aspirated into a sterile specimen trap (BALF Trap; Vigon, Ecouen, France) with 50 mmHg of negative pressure. This procedure was repeated with the head turned to the left, so that the right lung would be predominantly sampled. This procedure respects European Respiratory Society advices for pediatric and neonatal BAL[22]. During the procedure, the patient was never disconnected from the ventilator, the inspired oxygen fraction was 0.25 and there was no desaturation or bradycardia. All specimens were kept at +4 °C and tested within 24 h.

**Real-time polymerase chain reaction (RT-PCR)**. Viral RNA was extracted from 200 µL clinical samples with the NucliSENS® easyMag® (BioMérieux, Craponne, France) and eluted in 100 µL. The RealStar® SARS-CoV-2 RT-PCR Kit 1.0 (Altona Diagnostics GmbH, Hamburg, Germany) targeting the E gene (specific for lineage B-betacoronavirus) and the S gene (specific for SARS-CoV-2) was used following the manufacturer's recommendations ([https://altona-diagnostics.com/en/products/reagents-140/reagents/realstar-real-time-pcr-reagents/realstar-sars-cov-2-rt-pcr-kit-ruo.html](https://altona-diagnostics.com/en/products/reagents-140/reagents/realstar-real-time-pcr-reagents/realstar-sars-cov-2-rt-pcr-kit-ruo.html)). The assay includes a heterologous amplification system (internal positive control) to identify possible RT-PCR inhibition and to confirm the integrity of the reagents of the kit. The positive control is a SARS-CoV-2 culture supernatant provided by the kit manufacturer. Thermal cycling was performed at 55 °C for 20 min for reverse transcription, followed by 95 °C for 2 min and then 45 cycles of 95 °C for 15 s, 55 °C for 45 s, 72 °C for 15 s with an Applied Biosystems ViiA7 instrument (Applied Biosystems, Thermo Fisher, Waltham, MA, USA). A cycle threshold value less than 40 is interpreted as positive for SARS-CoV-2 RNA. Our technique resulted to have an extremely low limit of detection (LOD = 1200 cp/mL (12 cp/rxn)). Reproducibility and inter-assay agreement were 100% both for negative and for positive tests, against two other common techniques[23].

**Placental examination**. Placental sampling, gross and microscopic examination were performed according to the Amsterdam Consensus statement[24]. The placenta was fixed in 4% buffered formalin and samples were paraffin embedded. Staining methods performed on 3–5 µm thick sections were: haemalun eosin saffran, periodic acid schiff and Gomori-Grocott stains. Immunohistochemistry with peroxydase detection and hemalun counterstain was performed in a Leica Bond III automat using the Bond Polymer Refine Detection kit (Leica DS9800) after heat pretreatment at pH6 or 9 depending on the monoclonal antibodies tested: CD68 (Dako PG-M1, 1:200), CD163 (Leica 10D6, 1:200), CD20 (Dako L26, 1:400), CD3 (Dako F7.2.38, 1:50), CD5 (Novocastra 4C7, 1:50), CMV (Dako CCH2 + DDG9, 1:1), Parvo virus (AbcVs, Abc10-P038), SARS-CoV-2 (Abclonal, rabbit pAB, 2019-nCoV N Protein, 1:2400). Negative controls for SARS-CoV-2

immunohistochemistry were done: control of the polyclonal rabbit primary antibody, SARS-CoV-2 negative placental specimen with similar pre-analytic conditions of formalin fixation.

**Ethics declaration**. Written informed consent was obtained from the woman for the publication of this report. According to French regulation, institutional review board (IRB) approval is not required for case reports, provided that patients' written consent is obtained. The French Ethical Committee for the Research in Obstetrics and Gynecology reviewed the work and confirmed that the IRB approval was unnecessary. The case study was performed in agreement with principles of the Declaration of Helsinki and CARE guidelines[25].

**Reporting summary**. Further information on research design is available in the Nature Research Reporting Summary linked to this article.

## Data availability

All data generated or analyzed during this study are included in this published article.

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

## Author contributions

A.V. and C.V.F. managed the mother and performed the whole virological study, performed the literature search, prepared the figures, interpreted the data and wrote the manuscript draft. S.P. performed the pathological examination, prepared the figures and interpreted the data. V.Z. and C.S. performed and interpreted the neuroimaging. J.D.C. helped to manage the neonate and interpreted the whole clinical picture and laboratory tests. A.B. helped in literature search, data collection and interpretation and in the woman management. D.D.L. wrote the manuscript draft, managed the neonate, conceived the project and merged all the data. All authors critically reviewed the manuscript for important intellectual content and approved it in its final version.

## Competing interests

The authors declare no competing interests.
