## [Peer Review File · Nature Communications]

REVIEWER COMMENTS

Reviewer #1 (Remarks to the Author):

Vivanti et al. report transplacental transmission of SARS-CoV-2 infection. Viral RNA was detected in placenta, amniotic fluid, and baby blood. Since detection of the virus is the key to the conclusion, the authors should add results on virus isolation from these tissues. In addition, immunostaining of these tissues/cells with viral proteins will also support the conclusion.

Minor point:

Figures 2 and 4: arrows should be added to the figures, as indicated in the legends.

Reviewer #2 (Remarks to the Author):

Thank you for asking me to review this manuscript. There are some interesting findings which are unique; however, overall tone needs to be deemphasized to indicate “probable” congenital infection unless the following items are clarified further. Additionally, neurological findings need to be reported as an association for now.

Abstract:

1. Would suggest removing claims of “first time” from manuscript as there are other cases reported in literature and more reports are forthcoming.

Introduction:

1. First paragraph – again remove claims of “first time”.
2. What is category III fetal tracing and what is category II cesarean section – please clarify.
3. Was the cesarean section performed due to maternal cough? It says maternal respiratory status but not indicated what was her status from respiratory perspectives?
4. What was the reason for maternal admission for 6 days after delivery?

5. What was the reason for Apgar score going down from 4 to 2 if resuscitation was performed according to guidelines?
6. What were the reasons for POC echo and lung ultrasound examination?
7. What is the results of CSF RT PCR? What is detailed CSF report?
8. What is meant by placenta was extremely positive – was it swabs? Was it maternal side or fetal side?
9. How do authors explain signs of chronic intervillitis? If mother was infected only 5 days prior to birth?
10. What was trophoblastic involvement?
11. Was viral gene isolated from placental tissues?

Discussion:

1. Remove “first”.
2. There is a report of presence of viral genes in amniotic fluid, so this needs to be corrected.
<https://obgyn.onlinelibrary.wiley.com/doi/abs/10.1002/pd.5713>.
3. I would refrain from using term “vertical” or “Horizontal” as it should be more or less mechanism based rather than direction based.
4. BAL methods reference is author’s own paper where the method is not the investigation target. (reference 3 in online methods).
5. Before confirming the case as “confirmed” case there are several clarifications needed to be fulfilled
 - a. What is the false positive test rate of this testing? Above cycle threshold of 30, what is the false positive rate?
 - b. Crushed sample of placenta – suggest possibility of contamination by maternal blood – how can this be differentiated?
 - c. Newborn blood and maternal vaginal swab has very high cycle threshold (to the level which could be considered even negative) – please elaborate more.
 - d. Figure 3 has 3 Newborn NP swabs – which one is on which day?
 - e. Confirmed case would require clinical features of infection as well – for which clinical phenotype is still unknown and what neurological findings baby had could be an association with possible birth asphyxia picture (low Apgar, need for resuscitation, need for respiratory support) etc.
6. Emphasis is given on transplacental transmission – which indicate that it look like peripartum transmission. Is this what authors intend to say?

7. Before neurological findings are ascribed to possibility of association with SARS-CoV-2 further information is needed

a. What was baby's first blood gas and what recovery occurred in blood gas over time?

b. What was baby's first blood count?

c. What was baby's clinical neurological examination at birth and immediately thereafter?

d. What sedation/muscle relaxants baby received on first day?

e. Were virological studies conducted on CSF sample?

f. Please provide full CSF report for better interpretation.

8. Figure 4, panel A – arrow is not seen and how is it possible to see fibrin deposition macroscopically?

9. The placental histopathology revealed chronic histiocytic intervillitis, which is a chronic immunologic disorder associated with intrauterine growth retardation and miscarriage or stillbirth. How authors think it is associated with congenital infections especially viral infection. Where are the histiocytes located? Around the chorionic villi or elsewhere?

10. The significance of the cycle threshold (Ct) results for the PCR tests performed on the mother and child is not explained in the manuscript. First of all, for clinicians, it would be important to explain the inverse relationship of Ct number to viral load.

11. It will be useful in appendix to have table of investigations with cycle threshold values and an explanation that the viral loads (as estimated from Ct values) were correlate toetc.

12. Reference 16 is suggested for diagnosis of SARS-CoV-2 in neonates from China. Authors mention that their study meets this definition. Could you please clarify further. The paper indicates that baby should meet all four criteria; one of which includes "chest radiographs showing abnormalities, including unilateral or bilateral milled glass opacities". Please report any chest x-ray findings after intubation or soften that conclusion.

13. In the absence of any other organ involvement, too much emphasis is placed on MRI findings. This needs to be softened to an association.

14. What is inherent reliability of testing as indicated by manufacturer and field testing for this particular test and it need to be reported.

15. What treatment did baby receive? Did baby receive any antivirals?

16. What follow up data are available for baby and mother?

17. Would authors suggest change in timeline for testing of newborns born to mother who is SARS-CoV-2 positive, which currently is around 24 hours of age? based on their findings?

General comment:

1. Change all reference to isolation of “SARS-CoV-2 gene target isolation” by RT-PCR and not “virus isolation”.
2. Figure 1 – does not add much to paper and can be removed.
3. Table of laboratory findings may be more useful.

Reviewer #1

We thank the Reviewer for the thoughtful comment which was also shared by the Editor. As suggested, we have provided results of placental immunostaining for viral proteins (now added as Figure 5) and updated the main text and the online methods, accordingly. We feel that, thanks to this suggestion and the other details provided to Reviewer 2, we made a stronger case of transplacental transmission, the manuscript is much improved and we are very grateful to the Reviewer. Changes are provided in red font in the manuscript. We are willing to provide further clarification/amendments if needed.

1. Major point.

As suggested, we performed immunostaining and we provide results in the **main text, pag.5, 1st par** and in the **new figure 5, pag.16**. We also added all the viral loads for each analyzed specimen in **figure 3**.

All these data are consistent and support our conclusions. In fact, 1) maternal viremia occurred and the virus reached the placenta; 2) virus was found in placental tissue by immunostaining and the viral load in the placenta is much higher than elsewhere, that is, the virus is actively replicating in placental tissue with inflammatory reaction as seen by histological exam and immunohistochemistry; 3) neonatal viremia occurred following placental infection. Of note RT-PCR curves of NP swabs at 3 (while the baby was in full isolation in a negative pressure room) and 18 days of life are higher than that at the first day: this is also another confirmation that we observed an actual vertical infection, rather than a contamination.

Some of these considerations have been added in the main text, case study, end of pag.4-beginning of pag.5 and in discussion, end of pag.5-beginning of pag.6

2. Minor point:

Figures 2 and 4: arrows should be added to the figures, as indicated in the legends.

We are sorry, arrows were already there but, for a technical problem, they were not visible when the figure was embedded in the word file. Problem solved now.

Reviewer #2

We thank the Reviewer for the thoughtful comments. We followed all his/her suggestions and modified the manuscript accordingly.

We took the liberty to detail the diagnostic process, as expert neonatologists, to explain while the observed neurological manifestations cannot be ascribed to any other disorder than SARS-CoV-2 infection; we anyway refer to these symptoms as "associated" without a direct cause-effect relationship, as suggested. We also provided all the required details of both clinical care and virological testing. Moreover, as suggested by the Editor, we have provided results of placental immunostaining for viral proteins (now added as Figure 5) and updated the main text and the online methods, accordingly.

We feel that, thanks to all these, we made a stronger case of transplacental transmission, the manuscript is much improved and we are very grateful to the Reviewer! We anyway toned down and refrain to use sentences like "for the first time" as required.

Changes are provided in red font in the manuscript. We are willing to provide further clarification/amendments if needed.

1. Would suggest removing claims of "first time" from manuscript as there are other cases reported in literature and more reports are forthcoming.

Removed, as suggested

2. Introduction: First paragraph – again remove claims of "first time".

Done, as suggested

3. What is category III fetal tracing and what is category II cesarean section – please clarify.

According to the American College of Obstetrics and Gynecology guidelines (reference n.7 in the manuscript), the fetal heart rate (FHR) tracing is defined as category III it has either:

- Absent baseline FHR variability and any of the following:
 - Recurrent late decelerations
 - Recurrent variable decelerations
 - Bradycardia
- Sinusoidal pattern

The FHR of our cases showed tachycardia, absent baseline variability, absence of accelerations and recurrent late decelerations and, thus, it fulfilled the criteria for category III FHR tracings, which is strongly associated with adverse neonatal outcome.

According to the classification of the Royal College of Obstetricians-Gynecologists and Anesthetists (new reference 8 in the manuscript), a C-section is defined as category-II if it is an urgent delivery because of maternal or fetal compromise which is not immediately life-threatening (see classification below):

Figure 1. A classification relating the degree of urgency to the presence or absence of maternal or fetal compromise

Urgency	Definition	Category
Maternal or fetal compromise	Immediate threat to life of woman or fetus	1
	No immediate threat to life of woman or fetus	2
No maternal or fetal compromise	Requires early delivery	3
	At a time to suit the woman and maternity services	4

Therefore, we performed an urgent C-section based on the abnormal FHR tracing.

*We described better this issue and provided some of these data in the **main text, case study, pag.3** and in legend of **Figure 1 (pag.11)** which is an illustrative image taken from the FHR monitoring of our case, to help the average reader. We also added the reference to the classification of the Royal College of Obstetricians-Gynecologists and Anesthetists (now reference n.8 in the manuscript). We realize that these are very specialistic details that may be excessive or necessary according to the Reader's background. We are willing to add more detail if the Editor wish so.*

4. Was the cesarean section performed due to maternal cough? It says maternal respiratory status but not indicated what was her status from respiratory perspectives?

No. As explained above -see point 3- (and in the **main text, case study, pag.3**) the urgent C-section was indicated because of abnormal fetal heart rate tracing (although the woman presented with severe cough and expectoration, for 5 days before the delivery).

5. What was the reason for maternal admission for 6 days after delivery?

The mother remained hospitalized for six days after the delivery for surveillance of her clinical (particularly respiratory) conditions due to COVID19. Added in the **main text, pag.3, 1st par.**

6. What was the reason for Apgar score going down from 4 to 2 if resuscitation was performed according to guidelines?

The reason is related to the nature of Apgar score's items and the timing of the different resuscitation steps. **The observed reduction is absolutely nothing strange and often occur.** In detail:

1. The Apgar score is a classical tool evaluating 5 vital parameters (heart rate, respiratory activity, muscular tonus, skin color and grimace) giving 0-2 points for each item as shown in the table on the right: Our baby had an Apgar of 4 at 1 minute of life (in detail: heart rate=1, respiratory activity=1, skin color=1, muscular tonus=1, remaining items were coded zero). According to current guidelines (reference n.9 in the manuscript), non-invasive manual ventilation with face mask and T-piece resuscitator was started (with PEEP=5 cmH₂O and peak pressure 25 cmH₂O in room air, respiratory rate 40-45/min), while heart rate was monitored with ECG.

2. Our baby had an Apgar of 2 at 5 minutes (in detail: skin color=1, muscular tonus=1, remaining items were coded zero). This means that the non-invasive ventilation did not improve the situation, as the heart rate and the respiratory activity were no more evident. Thus, the neonatal fellow promptly intubated the baby (intubation verified by end-tidal CO₂ measurement) and started invasive ventilation at 6 minutes of life. As usual, the invasive ventilation was very efficacious in resuscitating the baby (non-invasive ventilation does not always work for a number of reasons, such as patency of upper airways, difficult patient's positioning, lack of synchronization, leaks reducing pressure delivery...).

APGAR Test Scoring		Score 0	Score 1	Score 2
Appearance				
	Blue all over	Blue only at extremities	No blue coloration	
Pulse	No pulse	<100 beats/min.	>100 beats/min.	
Grimace				
	No response to stimulation	Grimace or feeble cry when stimulated	Sneezing, coughing, or pulling away when stimulated	
Activity				
	No movement	Some movement	Active movement	
Respiration	No breathing	Weak, slow, or irregular breathing	Strong cry	

Thus, the Apgar at 5 minutes did not capture the effect of resuscitation as the main resuscitative maneuver (intubation and invasive ventilation) occurred at 6th minute. In fact, Apgar has been re-coded at 10 minutes of life and it was 7 (in detail: heart rate=2, respiratory activity=2, skin color=2, muscular tonus=1).

3. In its original formulation (still used worldwide) the Apgar score does not take any resuscitative maneuvers into consideration (although the American Academy of Pediatrics suggested to include negative points for each item depending on the type of resuscitative actions provided – AAP Position paper Pediatrics 2006; Rudiger M. NeoReview 2012). **In other words, the classical Apgar score describes the newborn conditions regardless of any medical intervention. Thus, the Apgar score itself does not describe correctly the effect of neonatal resuscitation even when the resuscitative interventions are provided at the same moment of Apgar calculation.**

*We better described the Apgar scores of our baby and the resuscitation timing and details in the **main text, cases study, pag.3, 2nd par** to help the average reader. We realize that these are very specialistic details that may be excessive or necessary according to the Reader's background. We are willing to add more detail if the Editor wish so.*

7. What were the reasons for POC echo and lung ultrasound examination?

According to our NICU clinical protocols, POC echo and lung ultrasound are routinely performed by attending/fellow neonatologists in all patients admitted to the NICU for cardio-respiratory compromise, suspected heart/lung malformation or for post-delivery room resuscitation care. **Thus, our protocols fully comply with the “International evidence-based guidelines on Point of Care Ultrasound (POCUS) for critically ill neonates and children issued by the POCUS Working Group of the European Society of Paediatric and Neonatal Intensive Care” (Singh Y et al. Crit Care 2020 - doi.org/10.1186/s13054-020-2787-9).**

In our case, this was obviously performed to detect any hemodynamic problem (impaired contractility, need for volume resuscitation, pulmonary pressure measurement) or respiratory disorders (post-resuscitation lung oedema, transient tachypnoea of the neonate, RDS, others...)

*A small explanation has been added in the **main text, case study, pag.3, last par.** and the international POCUS guidelines have been cited (**reference 10**).*

8. What is the results of CSF RT PCR? What is detailed CSF report?

Added as suggested (**main text, case study, pag.4, 2nd par.** and in the **new Table 1** reporting main laboratory findings) – see also below point 27,28 and 41

9. What is meant by placenta was extremely positive – was it swabs? Was it maternal side or fetal side?

In the placenta, the terminal villi (composed of capillaries (fetal blood), fetal macrophages and fibroblasts and surrounded by trophoblast cells) are surrounded by the intervillous space containing the maternal blood. So RT-PCR performed on placental tissue specimens unavoidably contain both maternal and fetal components.

However, there was no placental swab, but a real sample of placental tissue was taken from the chorionic (fetal) side, crushed and homogenized to be analyzed by RT-PCR, as explained in online methods, pag.1, par.1 “A sample of placental tissues was taken from the chorionic side and crushed in 400 mL of RNAase-DNAase-free water”. This ensure a fetal representative sampling.

Therefore, placental specimen used for RT-PCR techniques is collected from the chorionic (that is, fetal) side of the placenta, thus with predominance of fetal side. Moreover, the different viral load (viral load is extremely higher in the placenta than in maternal blood!) ensure that we did not observe a simple “contamination” but a true infection, as the virus replicates in the placental tissues.

*We added some of these considerations in **main text, case study, pag.4, last par** and in **Figure 3-Panel C** and in **Discussion, pag. 5-6***

10. How do authors explain signs of chronic intervillitis? If mother was infected only 5 days prior to birth?

Chronic intervillitis is not the only type of inflammation present in the placenta of our case. There is a polymorphonuclear cellular infiltrate composed of neutrophils, consisting with the acute phase of inflammation, associated with histiocytes appearing few days after the beginning of the process. As inflammation is a perpetual phenomenon it is common to observe an « active chronic inflammation » in placental tissues, such as here an « active chronic intervillitis ». Such phenomenon has been described in the placenta in case of viral infections, associated with recent infectious symptoms and is well known in feto-pathology. Of note, the mother was symptomatic for 5 days before the delivery but likely to be infected much earlier (impossible to know how long the period free from symptoms lasted).

We are willing to add some of these considerations in the main text if required by the Editor or the Reviewer.

11. What was trophoblastic involvement?

On morphological microscopic examination on HES stains, peri-villous trophoblastic cells (cyto- and syncytio-trophoblastic cells) showed no morphological alterations except in the territories of diffuse perivillous fibrin deposition with infarction. However, on immunohistochemistry, in the cytoplasm of the trophoblastic cells a strong positive signal was observed with the SARS-CoV-2 N-protein antibody.

*This, as suggested by the Editor, confirms the presence of the virus in these cells/tissue and has been added in **Fig.5** and in main text, case study, pag.5, 1st par and commented in **Discussion, pag.6, 1st par.***

12. Was viral gene isolated from placental tissues?

Yes, as shown in **figure 3 both E (fig.3A) and S (fig.3B)** viral genes were found in placental tissue. These were found in very high amount as indicated by number of PCR cycles and by the viral load that has now been added to this figure as **Panel C**. It is also reported in the **main text, case study, pag.4, last par** – See also point 9 above (RT-PCR was performed on predominantly fetal side of placenta).

13. Discussion: Remove “first”. Done, as suggested

14. There is a report of presence of viral genes in amniotic fluid, so this needs to be corrected. <https://obgyn.onlinelibrary.wiley.com/doi/abs/10.1002/pd.5713>.

Done, as suggested. This has also been added to the reference list. This manuscript does not reduce the interest of our work, since these authors (as other already cited) did not systematically test placenta, amniotic fluid and mother/newborn blood in every mother-infant pair.

15. I would refrain from using term “vertical” or “Horizontal” as it should be more or less mechanism based rather than direction based.

The Reviewer is totally right. Corrected throughout the manuscript

16. BAL methods reference is author’s own paper where the method is not the investigation target. (reference 3 in online methods).

The Reviewer is right, as that paper target was not the BAL method validation. However, it is an ideal citation as it describes in detail the neonatal BAL technique used in our unit. We will describe it more in detail directly in the main text or in the online methods if the Editor or the Reviewer wish so.

The technique in use is also the commonest one in neonatal critical care and is used in the majority of research projects in the field and able to retrieve more fluid from alveoli and distal airways than other simpler techniques (Dargaville P. Am J Resp Crit Care Med 1999 - DOI:10.1164/ajrccm.160.3.9811048). We shall also remember that no gold standard technique for neonatal BAL exists (as acknowledged by the European Respiratory Society guidelines on pediatric BAL – reference n.4 of online methods), thus the most important thing is to describe which technique is used and that the technique would be correctly standardized and performed.

17. Before confirming the case as “confirmed” case there are several clarifications needed to be fulfilled

What is the false positive test rate of this testing? Above cycle threshold of 30, what is the false positive rate?

PCR is highly specific. False positive results are extremely uncommon, because negative controls are included in each run as provided by the kit manufacturer.

This is confirmed by the curve (dark orange line) of the positive control which is a SARS-CoV-2 culture supernatant provided by the kit manufacturer (details in the online methods, pag.2, 2nd par): the positive control is also beyond 30 cycles and it cannot be a false positive by definition.

Anyway, the only samples with cycles above 30 are the vaginal swab (but the baby has been delivered by C-section), the amniotic fluid (but this is slightly above 30 and the placenta is way more positive) and the newborn blood (but the baby had several other positive samples demonstrating that he was actually infected – see points n.12-18 and 20)

Going more deeply in this area, the false positive rate of our RT-PCR technique has been recently estimated to be within 0.8 and 4% of cases (Cohen AN et al, posted on MedRxiv on May 1, 2020 - doi.org/10.1101/2020.04.26.20080911), but these low rates have an impact only when testing is done when prevalence is low.

Our case was observed in the worst moment of the COVID19 outbreak, in the Parisian area (Ile de France) which is amongst the more hardly hit in Europe

(see <https://www.santepubliquefrance.fr/maladies-et-traumatismes/maladies-et-infections-respiratoires/infection-a-coronavirus/articles/infection-au-nouveau-coronavirus-sars-cov-2-covid-19-france-et-monde#block-238896>)

and in a case presenting other 10 extremely positive samples (well below 30 cycles) and with a high suspicion index (clinical symptoms typical of COVID for both the mother and the neonate).

The US Center for Disease Control and Prevention (<https://www.cdc.gov/sars/guidance/f-lab/assays.html>) states that: 1) the most common cause of false-positive results is contamination. 2) Liberal use of negative control samples in each assay and a well-designed plan for confirmatory testing (with more than one gene) can help ensure that laboratory contamination is detected. 3) Testing should be limited to patients with a high index of suspicion for having COVID19.

*We do respect all these three conditions (we have negative and positive controls; all samples are positive for two viral genes; we have a lot of consistently positive different specimens from our case; there is a high clinical suspicion index) as described both in the **main text (case study)** and in the **online methods, 2nd par**, thus we can consider the possibility of false positive virtually zero. We added some of these considerations in **Discussion, pag.5, 3^d par**.*

18. Crushed sample of placenta – suggest possibility of contamination by maternal blood – how can this be differentiated?

As described above (points n.9 and 11 and point 1 of Reviewer 1) we provided enough data to exclude the simple contamination:

- 1) placenta was sampled mainly on the chorionic (that is, fetal) side;
- 2) viral load in the placenta is extremely higher than in maternal blood, thus there is an active viral replication in placental tissues;
- 3) immunostaining for viral proteins confirms the presence of the virus in placental cells;
- 4) neonatal viremia was observed and there was no other likely transmission route (as amniotic membranes were intact and the baby was delivered by C-section).

19. Newborn blood and maternal vaginal swab has very high cycle threshold (to the level which could be considered even negative) – please elaborate more.

RT-PCR in newborn blood and maternal vaginal were not negative since they were below the cycle threshold value of 40 as described in the **online methods, beginning of pag.2**. There are several reasons to consider this as positive:

1. **This threshold (40) is internationally used for RT-PCR in virological studies and particularly to detect SARS-CoV-2 infection in all recent papers on the subject** (Wang W et al, JAMA 2020 - doi:10.1001/jama.2020.3786; Corman VM et al, Euro Surveill 2020 - doi: 10.2807/1560; Daniel KWC et al. Clin Chem 2020 - doi: 10.1093/clinchem/hvaa029, amongst the others...). We feel it is important to keep using the same threshold for consistency.
2. Some Chinese colleagues proposed on the WHO Bulletin (Lan PT, et al. Bull World Health Organ. E-pub: 20 April 2020. doi: <http://dx.doi.org/10.2471/BLT.20.259630>) a standardized technique for RT-PCR considering as cycle threshold a value less than 36. **Even if we consider this threshold, all our samples are positive, with the exception of the vaginal swabs. However, this is irrelevant since the baby was delivered by C-section with intact amniotic membranes, so intrapartum transmission during a vaginal delivery did not happen.**
3. **According to the European Centre for Disease Control (ECDC - <https://www.ecdc.europa.eu/en/all-topics-z/coronavirus/threats-and-outbreaks/covid-19/laboratory-support/questions>), a high CT value (e.g. > 35) for the E-gene RT-PCR should be confirmed by a second gene target (to exclude a contamination) and all our samples from both the mother and the neonate were positive for two genes (the “E” and the “S” gene of SARS-CoV-2). Thus, our results were confirmed and interpreted according to ECDC guidelines.**

*Some of these considerations have been added in the **main text, discussion, pag.5, 3^d par***

20. Figure 3 has 3 Newborn NP swabs – which one is on which day?

NP swabs at 1, 3 and 18 day of life are represented by the light orange, grey and green curves, respectively. This has been now specified in Figure 3 legend. Of note RT-PCR curves of NP swabs at 3 and 18 day of life are higher than that at the first day (while the baby was in full isolation in a negative pressure room): this is also another confirmation that we observed an actual vertical infection, rather than a contamination.

*Some of these considerations (together with those about the viral loads) have been added in the **main text, discussion, beginning of pag.6***

21. Confirmed case would require clinical features of infection as well – for which clinical phenotype is still unknown and what neurological findings baby had could be an association with possible birth asphyxia picture (low Apgar, need for resuscitation, need for respiratory support) etc.

We respectfully disagree. Some of us are neonatologists with long experience and academic career (VZS and DLD), we can testify that the neurological findings of our baby are not linked to any other possible cause known so far. In detail the following were excluded:

- **Other infections (as LCR, blood and BAL were sterile for other bacteria, microbes and viruses).**
- **Consequences of perinatal asphyxia (since the baby was efficaciously resuscitated as proven by the first blood gas analysis showed normal pH and lactate. Moreover, Sarnat score, echocardiogram, blood test (including troponin, liver and kidney function were totally normal). Given these data the baby does NOT qualify 1) for the definition of significant perinatal asphyxia according to the American College of Obstetricians and Gynaecologists and the American Academy of Paediatrics (American College of Obstetricians and Gynecologists (ACOG). Neonatal Encephalopathy and Cerebral Palsy: Executive Summary. Obstet Gynecol 2004), nor for 2) nor for its hypothermic treatment (American Academy of Pediatrics-Committee on Fetus and Newborn, Pediatrics 2014 - <https://doi.org/10.1542/peds.2014-0899>).**
- **Metabolic diseases (as there was no acidosis, any metabolic derangement and no suspected clinical symptoms until almost 2 months of life).**

Moreover, the observed neurological manifestations are totally consistent with those seen by colleagues in adult critical care (reference n.16 and other more recent that have highlighted neurological involvement in COVID19 (Calcagno N et al, *Neurol Sci* 2020 - doi.org/10.1007/s10072-020-04447-w; Asadi-Pooya AA et al. *J Neurol Sci* 2020 - [doi: 10.1016/j.jns.2020.116832](https://doi.org/10.1016/j.jns.2020.116832); Li Z et al, *Front Med* 2020 - [doi: 10.1007/s11684-020-0786-5](https://doi.org/10.1007/s11684-020-0786-5); Zhou Y et al, *Stroke Vasc Neurol* 2020 - [doi: 10.1136/svn-2020-000398](https://doi.org/10.1136/svn-2020-000398), amongst the others). Particularly, neurological complications due to inflammatory response rather than to the direct viral infection are not so rare and the Neurocritical Care Society launched an international data collection to study this issue (Needham EJ et al, *Neurocrit Care* 2020 - [doi: 10.1007/s12028-020-00978-4](https://doi.org/10.1007/s12028-020-00978-4)). **Finally, another case of neurological manifestations in a neonate born to mother with COVID19 has been recently published (now cited as reference n.13 in the manuscript).**

We added some of these clinical reasoning in the main text, case study, pag.4 and in Discussion, end of pag.6- beginning of pag.7 also citing the statement of the Neurocritical Care Society (reference n.20 in the manuscript)

We anyway described these neurological manifestations as only “associated” to the SARS-CoV-2 infection, as suggested (that is, we refrained to indicate a direct causal link).

22. Emphasis is given on transplacental transmission – which indicate that it looks like peripartum transmission. Is this what authors intend to say?

Not actually, we focused on transplacental transmission which is a maternal-fetal (congenital) transmission occurred during the pregnancy. This is totally different from the intra-partum transmission which is an infection acquired during vaginal birth and also different from the post(per)partum transmission which means an infection transmitted from the mother during mother/infant bonding or newborn care or breastfeeding and/or transmitted from the environment.

The differences between these transmission routes are well described in the International Classification of for SARS-CoV-2 infection in pregnant women, fetuses, and neonates, that we apply in clinical care and that is cited in the manuscript (reference n.14). According to this classification criteria, our case is a transplacental (congenital) infection in a live born neonate, as we described in Discussion, pag.5, 3rd par.

We focused on transplacental transmission (i.e.: congenital) transmission occurred during the pregnancy, because this is what is really new and never clearly demonstrated so far.

23. Before neurological findings are ascribed to possibility of association with SARS-CoV-2 further information is needed What was baby's first blood gas and what recovery occurred in blood gas over time?

As we described above (point n.21), blood gas analyses were normal and **the baby did not qualify neither for the definition of significant perinatal asphyxia (American College of Obstetricians and Gynecologists (ACOG). Neonatal Encephalopathy and Cerebral Palsy: Executive Summary. Obstet Gynecol 2004), nor for its hypothermic treatment (American Academy of Pediatrics-Committee on Fetus and Newborn, Pediatrics 2014 - <https://doi.org/10.1542/peds.2014-0899>).**

We have added a specific table (Table 1-pag.13) resuming the main laboratory findings, as suggested. Here you find the evolution of patient's blood gas analyses (all performed on arterialized blood drawn from warmed heel prick):

	DOL1	DOL2	DOL2	DOL3
pH	7,27	7,38	7,39	7,34
pCO2 mmHg	41	41	41	47
pO2 mmHg	41	40	27	30
BE mmol/L	-7,8	-1,4	-0,4	-1
Lactates mmol/l	7	1,3	1,5	1,5

24. What was baby's first blood count?

All newborn cell blood counts were perfectly normal as stated in the main text.

We have added a specific table (Table 1-pag.13) resuming the main laboratory findings.

Here you find the evolution of his blood counts (please note that all these values are perfectly normal for the reference in newborn infants):

	DOL1	DOL1	DOL2
Leucocytes /L	10.32x10 ⁹	9,09x10 ⁹	6,97x10 ⁹
Red cells /L	4.54x10 ¹²	4,98x10 ¹²	4,84x10 ¹²
Hb g/dL	13,9	15,2	14,7
Hematocrit %	41,6	43	41,4
VGM fL	91,6	86,3	85,5
TCMH pg	30,6	30,5	30,4
CCMH g/dL	33,4	35,3	35,5
IDR %	15,9	14,6	14,7
Platelets /L	339x10 ⁹	328x10 ⁹	319x10 ⁹
VMP fL	9	8,8	9,2
PNN %	38,5	38,1	39,8
PNN /L	3,97x10 ⁹	3,46x10 ⁹	2,78x10 ⁹
PNE %	3,3	0,4	3
PNE /L	0,34x10 ⁹	0,04x10 ⁹	0,21x10 ⁹
PNB %	1,4	0,7	0,6
PNB /L	0,14x10 ⁹	0,06x10 ⁹	0,04x10 ⁹
Lymphocytes %	42,5	51,7	43,8
Lymphocytes /L	4,39x10 ⁹	4,7x10 ⁹	3,05x10 ⁹
Monocytes %	14,3	9,1	12,8
Monocytes /L	1,48x10 ⁹	0,83x10 ⁹	0,89x10 ⁹
Immature PNN %	4,9	2,2	0,6
erythroblastes %	4,9	2,4	3,2
reticulocytes %	3,04	3,2	3,39
réticulocytes /L	138x10 ⁹	159,4x10 ⁹	164,1x10 ⁹

25. What was baby's clinical neurological examination at birth and immediately thereafter?

As we already explained in the **main text, case study, end of pag.3**, the neurological exam at the birth and in the first hours, classified according to Sarnat score was totally normal and this is why the baby was almost immediately extubated upon NICU admission.

Sarnat score is the classical diagnostic tool internationally used for the evaluation of neurological conditions of neonates after delivery room resuscitation (Sarnat H et al, Arch Neurol 1976 - 10.1001/archneur.1976.00500100030012)

26. What sedation/muscle relaxants baby received on first day?

Absolutely nothing. The neonate did not receive any sedation/analgesia. This is our internal protocol to achieve a more detailed neurological examination in babies after delivery room resuscitation and is also recommended by the American Academy of Pediatrics-Committee on Fetus and Newborn, Pediatrics 2014 - <https://doi.org/10.1542/peds.2014-0899> for the "virgin" background evaluation of babies potentially candidate to therapeutic hypothermia for perinatal asphyxia-induced encephalopathy).

We better specified this in the **main text, case study, end of pag.3**

27. Were virological studies conducted on CSF sample?

Yes, as we already wrote in the **main text, case study, pag.4, 2nd par**, RT-PCR was done on two LCR obtained on DOL3 (that is when neurological examination started) and DOL5 and both were negative for SARS-CoV-2. Conversely, the first showed signs of inflammation (300 leukocytes/mm³ and slightly raised proteins (1.49 g/L)). We have now better detailed this in the **main text, case study, pag.4, 2nd par**. See also above point 8

28. Please provide full CSF report for better interpretation.

We have now better detailed this in the **main text, case study, pag.4, 2nd par** and added a specific table (Table 1-pag13), resuming the main laboratory findings. See also points n.8, 23 and 24 above.

Please find below all the LCR test results:

	DOL3	DOL5
protéins g/L	1,49	1,24
glucose mmol/L	2,9	2,4
HSV1 DNA	neg	neg
HSV2 DNA	neg	neg
Entérovirus RNA	neg	neg
COVID19 RNA	neg	neg
leucocytes /mm3	300	11
other cells%	0	0
red cells /mm3	2000	4800
Gram staining	neg	neg
Culture	neg	neg

29. Figure 4, panel A – arrow is not seen and how is it possible to see fibrin deposition macroscopically?

We are sorry, arrows were already there but, for a technical problem, they were not visible when the figure was embedded in the word file. Problem solved now.

Massive perivillous fibrin deposits are classically visible on gross examination of the cut sections of the placenta as irregular strands or masses of pale yellow-white induration and so it is in our case. We added this description in the **legend of Figure 4**.

30. The placental histopathology revealed chronic histiocytic intervillitis, which is a chronic immunologic disorder associated with intrauterine growth retardation and miscarriage or stillbirth. How authors think it is associated with congenital infections especially viral infection. Where are the histiocytes located? Around the chorionic villi or elsewhere? Intervillitis, is also commonly seen in different conditions such as viral infections (and not only in immunologic disorders), as already reported above (points n.9-10). Histiocytes, as well as neutrophils, are present in the intervillous space and are maternal inflammatory cells. There was no sign of villitis either acute or chronic in our case. Anyway, we did not intend to indicate intervillitis as a direct effect of SARS-CoV-2 infection but just to report the associate findings. We are willing to add some of these considerations in the text if required by the Editor or the Reviewer.

31. The significance of the cycle threshold (Ct) results for the PCR tests performed on the mother and child is not explained in the manuscript. First of all, for clinicians, it would be important to explain the inverse relationship of Ct number to viral load.

We thank the Reviewer and we have added a brief explanation in **Figure 3 legend**.

32. It will be useful in appendix to have table of investigations with cycle threshold values and an explanation that the viral loads (as estimated from Ct values) were correlate toetc.

We added all the viral loads as a table (Panel C) in the Figure 3 and explained this in the figure legend. The main results (viral load in placenta, amniotic fluid and maternal and neonatal blood) are also highlighted in the main text, case study, end of pag.4

This allows to understand that 1) maternal viremia occurred and the virus reached the placenta; 2) viral load in the placenta is much higher than elsewhere, that is the virus is actively replicating in placental tissue with inflammatory reaction as seen by histological exam and immunohistochemistry; 3) neonatal viremia occurred following placental infection. Some of these considerations have been added in the main text, discussion, pag.6, 1st par.

33. Reference 16 is suggested for diagnosis of SARS-CoV-2 in neonates from China. Authors mention that their study meets this definition. Could you please clarify further. The paper indicates that baby should meet all four criteria; one of which includes “chest radiographs showing abnormalities, including unilateral or bilateral milled glass opacities”. Please report any chest x-ray findings after intubation or soften that conclusion.

We thank the Reviewer as this was an inaccuracy from our side. That Chinese manuscript is quite “old” in this quickly evolutive pandemic situation: it was published in February, when extrapulmonary manifestations of COVID19 were not well known. Thus, the criteria reported by these Chinese colleagues are not adapted to our case as our baby did not have any respiratory disorder. We deleted this part and also the reference, as suggested.

34. In the absence of any other organ involvement, too much emphasis is placed on MRI findings. This needs to be softened to an association.

This has been softened and indicated only as an association through all the manuscript.

35. What is inherent reliability of testing as indicated by manufacturer and field testing for this particular test and it need to be reported.

The RT-PCR we used is extremely accurate and it has been recently subjected to a comparative analysis versus the United States CDC method and another commercial kit (Uhteg K et al, J Med Virol 2020 - doi:10.1016/j.jcv.2020.104384).

In this study, our kit resulted to have an extremely low limit of detection (LOD = 1,200 cp/ mL (12 cp/ rxn)). The inter-assay agreement was 100% both for negative and for positive tests, against both the other two techniques (see below).

TABLE 2: Agreement between the RealStar® SARS-CoV-2 and the CDC COVID-19 RT-PCR assays

Sample #	RealStar® SARS-CoV-2		CDC COVID-19		
	B-βCoV ^a	SARS-CoV-2 ^b	N1	N2	RP ^c
1	23.60	22.13	24.01	24.39	23.21
2	19.31	17.47	21.22	20.90	22.38
3	25.82	24.02	27.59	27.93	25.10
4	20.54	19.03	22.28	22.34	22.69
5	21.46	19.80	22.39	23.22	24.17
6	19.29	18.17	19.50	19.80	23.99
7	22.05	20.57	23.06	23.50	24.59
8	22.37	21.63	21.76	22.10	24.45
9	19.90	19.15	20.59	20.87	25.98
10	18.76	18.38	20.06	19.84	25.78
11	15.42	15.27	15.32	15.45	23.54
12	20.19	20.54	20.55	20.74	25.41
13	22.37	21.63	21.76	22.10	24.45
14	19.90	19.15	20.59	20.87	25.98
15	18.76	18.38	20.06	19.84	25.78
16	15.42	15.27	15.32	15.45	23.54
17	20.19	20.54	20.55	20.74	25.41
18	20.03	19.28	19.92	20.37	20.82
19	18.80	18.39	16.78	16.57	17.52
20	29.03	28.22	28.94	29.06	29

^a E gene

^b S gene

^c human RNase gene

TABLE 3: Agreement between the RealStar® SARS-CoV-2 and the ePlex® SARS-CoV-2 assays

Sample #	RealStar® SARS-CoV-2		ePlex® SARS-CoV-2
	B-βCoV ^a	SARS-CoV-2 ^b	
1	16.35	15.35	Positive
2	18.74	17.66	Positive
3	18.54	17.02	Positive
4	16.09	15.58	Positive
5	30.21	28.83	Positive
6	22.93	22.99	Positive
7	22.05	20.57	Positive
8	21.46	19.8	Positive
9	19.29	18.17	Positive
10	25.82	24.02	Positive
11	20.54	19.03	Positive
12	22.05	20.57	Positive
13	34.47	32.49	Positive

^a E gene

^b S gene

Reproducibility was also always 100% (see below).

TABLE 6: Reproducibility of the ePlex® SARS-CoV-2 Assay

Run	cp/rxn ^a	# reps	No. positive (% pos)	Bays used
1	2400	3	3/3 (100%)	A1-3
	240	6	6/6 (100%)	A1-6
	120	6	6/6 (100%)	A1-6
2	120	4	4/4 (100%)	A1-4
3	120	6	6/6 (100%)	A1-6

^a 200 µL per reaction

As the authors wrote: “The RealStar® SARS-CoV-2 assay [our test NoA] was the first assay validated and implemented at Johns Hopkins Hospital. To assess the range of viral loads detected by this assay since its implementation, we examined the Ct values of a subset of the initial positive specimens over time (Figure 1 – see below). Initially, diagnosed cases after the implementation of the assay had Ct values below 25. However, with increasing number of positive cases [with the progression of epidemics and high suspicion index], we detected a wide range of Ct values from less than 15 to more than 40” (see below).

[This means that, according to the US Center for Disease and Control guidelines \(https://www.cdc.gov/sars/guidance/f-lab/assays.html\), in the middle of epidemic peak, positive RT-PCR results, also with relatively high number of cycles, are true positive if there are clinical symptoms \(high suspicion index\), as in our case. See also above points n.17 and 19](https://www.cdc.gov/sars/guidance/f-lab/assays.html)

This also answer to the question (point n.17) regarding the accuracy of results between 30 and 35 cycles. We added some of these considerations and the aforementioned reference in the online methods, pag.2, 1st par.

36. What treatment did baby receive? Did baby receive any antivirals?

No, the baby did not receive antivirals nor any other specific treatment. This is now specified in **main text, case study, pag.4, 3rd par.**

37. What follow up data are available for baby and mother?

The baby is currently at almost two months of follow up, he has been re-examined and underwent a second MRI: the situation is slowly improving. This is now specified in **main text, case study, pag.4, 3rd par.**
The mother is fine and had no other problems.

38. Would authors suggest change in timeline for testing of newborns born to mother who is SARS-CoV-2 positive, which currently is around 24 hours of age? based on their findings?

The International classification of SARS-CoV-2 infection in pregnant women, fetuses, and neonates (Shah PS et al, Acta Obstet Gynecol Scand 2020 - DOI: 10.1111/aogs.13870, Reference n.14 in the manuscript) requires to test the newborn blood within 12h of life (viremia or presence of antibodies is within the diagnostic criteria of congenital infection), but also to repeat the nasopharyngeal swab at 24-48h or >48h of postnatal life as later positivity is a criterion for the diagnosis of intrapartum acquired or postnatal infection, respectively.

We suggest to follow this classification which is complex but useful to better understand the pathobiology of perinatal SARS-CoV-2 transmission. We actually followed it and did more investigations as needed to well characterize the case which is unique as the Reviewer acknowledged.

We added our suggestion to follow this classification, in the main text, discussion, pag.5, 3rd par and you can find the criteria of this classification below:

Congenital infection in live born neonate

Clinical features of infection in newborn and mother with SARS-CoV-2 infection	Confirmed	Detection of the virus by PCR in umbilical cord blood ^b or neonatal blood collected within first 12 hours of birth or amniotic fluid collected prior to rupture of membrane ^c
	Probable	Detection of the virus by PCR in nasopharyngeal swab at birth (collected after cleaning baby) AND placental swab from fetal side of placenta in a neonate born via cesarean section before rupture of membrane or placental tissue
	Possible ^a	No detection of the virus by PCR in nasopharyngeal swab at birth (collected after cleaning baby) BUT presence of anti-SARS-CoV-2 IgM antibodies in umbilical cord blood or neonatal blood collected within first 12 hours of birth or placental tissue
	Unlikely	No detection of the virus by PCR in nasopharyngeal swab at birth (collected after cleaning baby) or umbilical cord blood, or neonatal blood collected within first 12 hours of birth or amniotic fluid AND antibody testing not done
	Not infected	No detection of the virus by PCR in nasopharyngeal swab at birth (collected after cleaning baby) or umbilical cord blood, or neonatal blood collected within first 12 hours of birth or amniotic fluid AND no anti-SARS-CoV-2 IgM in umbilical cord blood or neonatal blood collected within first 12 hours of birth
No clinical features of infection in newborn and mother with SARS-CoV-2 infection	Confirmed	Detection of the virus by PCR in cord blood ^b or neonatal blood collected within first 12 hours of birth
	Probable	Detection of the virus by PCR in amniotic fluid collected prior to rupture of membrane but no detection in umbilical cord blood or neonatal blood collected within first 12 hours of birth
	Possible	Presence of anti-SARS-CoV-2 IgM in umbilical cord blood or detection of the virus by PCR in placental tissue but no detection of the virus by PCR in umbilical cord blood or neonatal blood collected within first 12 hours of birth or amniotic fluid
	Unlikely	No detection of the virus by PCR in cord blood or neonatal blood collected within first 12 hours of birth or amniotic fluid collected prior to rupture of membrane ^c AND serology not done
	Not infected	No detection of the virus by PCR in cord blood or neonatal blood collected within first 12 hours of birth or amniotic fluid collected prior to rupture of membrane ^c AND no anti-SARS-CoV-2 IgM in cord blood

(Continues)

Patient	Category	Case Definition
Neonatal infection acquired intrapartum		
Clinical features of infection in newborn and mother with SARS-CoV-2 infection	Confirmed	Detection of the virus by PCR in nasopharyngeal swab at birth (collected after cleaning the baby) AND at 24-48 hours of age AND alternate explanation for clinical features excluded
	Probable	Detection of the virus by PCR in nasopharyngeal swab at birth (collected after cleaning baby) but not at 24-48 hours of age AND alternate explanation for clinical features excluded
	Possible	No detection of the virus by PCR in nasopharyngeal swab at birth AND detection of the virus by PCR in any of maternal vaginal/placental/cord/skin swab at birth AND alternate explanation for clinical features excluded
	Unlikely	No detection of the virus by PCR in nasopharyngeal swab at birth (collected after cleaning baby) OR in any of maternal vaginal/placental/cord/neonatal nasopharyngeal/skin swab at birth AND alternate explanation for clinical features not identified
	Not infected	No detection of the virus by PCR in nasopharyngeal swab at birth (collected after cleaning baby) OR in any of maternal vaginal/placental/cord/neonatal nasopharyngeal/skin swab at birth AND alternate explanation for clinical features identified
No clinical features of infection in newborn and mother with SARS-CoV-2 infection	Confirmed	Detection of the virus by PCR in nasopharyngeal swab at birth (collected after cleaning the baby) AND at 24-48 hours of age
	Possible	Detection of the virus by PCR in nasopharyngeal swab at birth (collected after cleaning the baby) AND not at 24-48 hours
	Not infected	No detection of the virus by PCR in nasopharyngeal swab at birth AND no detection of the virus by PCR in any of vaginal swab in mother/placental swab/skin/cord swab at birth
Neonatal infection acquired postpartum		
Clinical features of infection in newborn at ≥48 hours age (parent or caregiver may or may not have SARS-CoV-2 infection or were not tested)	Confirmed	Detection of the virus by PCR in nasopharyngeal/rectal swab at ≥48 hours of birth in a neonate whose respiratory sample tested negative by PCR at birth
	Probable	Detection of the virus by PCR in nasopharyngeal/rectal swab at ≥48 hours of birth in a neonate who was not tested at birth
	Not infected ^a	No detection of the virus by PCR in nasopharyngeal/rectal swab at ≥48 hours of birth and other cause identified

This system is for maternal SARS-CoV-2 infection diagnosed prenatally or within 2-3 weeks of birth.

Category definitions: Confirmed, Strong evidence of infection with confirmatory microbiology; Probable, Strong evidence of infection but confirmatory microbiology lacking; Possible, Evidence suggestive of infection but incomplete; Unlikely, Little support for diagnosis but infection cannot be ruled out; Not infected, No evidence of infection.

Abbreviations: IgM, immunoglobulin M; PCR, polymerase chain reaction.

^aIn highly suspicious cases, repeat sample may be needed due to test limitations.

^bCollected using sterile precaution and thorough cleaning of cord.

^cIncludes sample taken at cesarean section performed before rupture of membranes.

39. General comment: Change all reference to isolation of “SARS-CoV-2 gene target isolation” by RT-PCR and not “virus isolation”. Corrected, as suggested

40. Figure 1 – does not add much to paper and can be removed.

We prefer to keep this figure for now to better describe the case from the beginning. However, we are willing to delete it if this is considered necessary by the Reviewer or the Editor

41. Table of laboratory findings may be more useful.

Added, as suggested (now as **Table 1-pag.13**).

REVIEWERS' COMMENTS:

Reviewer #1 (Remarks to the Author):

The authors have adequately addressed this reviewer's comments.

Reviewer #2 (Remarks to the Author):

thank you for revision of the manuscript. it has improved significantly.

further comments to address:

Abstract:

1. Last sentence – replace “consistent” to “similar”.

Text:

1. The age in hours at first blood gas analyses will be useful to know as it can help to satisfy that neonate was not asphyxiated.

2. Reference to “placenta was extremely positive” should be changed – there is nothing like “extremely” or not extremely – it was positive.

3. Presence of viral gene in placenta could still be contaminated by blood. Need to soften this finding.

Discussion:

1. Paragraph 1 will need to be revised based on one published case with similar placental findings. <https://www.cmaj.ca/content/early/2020/05/14/cmaj.200821>

2. Suggest to remove the concept of “viral replication” in placenta as it is possible that the high viral load in placenta may just reflect stagnation of blood and thus virus from maternal circulation. There is no evidence of replication based on the investigations performed. Inflammatory findings do not mean replication.

REVIEWERS' COMMENTS:

Reviewer #1 (Remarks to the Author):

The authors have adequately addressed this reviewer's comments.

We thank the reviewer for his/her thoughtful comments. The manuscript has greatly improved thanks to his/her work!

Please find below our point-to point reply to your last comments. The manuscript has been changed according to your comments and those from the Editor. Changes are provided in blue light font through the manuscript. It has also ben rearranged according to Journal style.

Reviewer #2 (Remarks to the Author):

thank you for revision of the manuscript. it has improved significantly.

We thank the reviewer for his/her thoughtful comments. The manuscript has greatly improved thanks to his/her work!

Please find below our point-to point reply to your last comments. The manuscript has been changed according to your comments and those from the Editor. Changes are provided in blue light font through the manuscript. It has also ben rearranged according to Journal style.

further comments to address:

Abstract:

1. Last sentence – replace “consistent” to “similar”. **Done.**

Text:

1. The age in hours at first blood gas analyses will be useful to know as it can help to satisfy that neonate was not asphyxiated. **Added, in Results, Pag.4, (this was at 1.5h of life).**

2. Reference to “placenta was extremely positive” should be changed – there is nothing like “extremely” or not extremely – it was positive. **Corrected, as suggested.**

3. Presence of viral gene in placenta could still be contaminated by blood. Need to soften this finding. **Toned down removing “extremely” and also removing the concept of “active replication” (see below)**

Discussion:

1. Paragraph 1 will need to be revised based on one published case with similar placental

findings. <https://www.cmaj.ca/content/early/2020/05/14/cmaj.200821>

Thanks for this. Updated citing this new reference.

2. Suggest to remove the concept of “viral replication” in placenta as it is possible that the high viral load in placenta may just reflect stagnation of blood and thus virus from maternal circulation. There is no evidence of replication based on the investigations performed. Inflammatory findings do not mean replication. **The Reviewer is totally right. We have removed and rephrased.**